# PESSA: A web tool for pathway enrichment score-based survival analysis in cancer

**Hong Yang**[1,2‡], **Ying Shi**[1,3‡], **Anqi Lin**[1‡], **Chang Qi**[4], **Zaoqu Liu**[5,6], **Quan Cheng**[7,8]*, **Kai Miao**[9,10]*, **Jian Zhang**[1]*, **Peng Luo**[1]*

**1** Department of Oncology, Zhujiang Hospital, Southern Medical University, Haizhu District, Guangzhou, Guangdong, China, **2** The First School of Clinical Medicine, Southern Medical University, Baiyun District, Guangzhou, Guangdong, China, **3** The Second School of Clinical Medicine, Southern Medical University, Baiyun District, Guangzhou, Guangdong, China, **4** Institute of Logic and Computation, TU Wien, Austria, **5** State Key Laboratory of Proteomics, Beijing Proteome Research Center, National Center for Protein Sciences (Beijing), Beijing Institute of Lifeomics, Beijing, China, **6** State Key Laboratory of Medical Molecular Biology, Institute of Basic Medical Sciences, Chinese Academy of Medical Sciences, Department of Pathophysiology, Peking Union Medical College, Beijing, China, **7** Department of Neurosurgery, Xiangya Hospital, Central South University, Changsha, Hunan, China, **8** National Clinical Research Center for Geriatric Disorders, Xiangya Hospital, Central South University, Changsha, China, **9** Cancer Centre and Institute of Translational Medicine, Faculty of Health Sciences, University of Macau, Macau SAR, China, **10** MoE Frontiers Science Center for Precision Oncology, University of Macau, Macau SAR, China

‡ These authors contributed equally to this work and share first authorship.
* chengquan@csu.edu.cn (QC); kaimiao@um.edu.mo (KM); zhangjian@i.smu.edu.cn (JZ); luopeng@smu.edu.cn (PL)

**Data Availability Statement:** The datasets supporting the conclusions of this article are included within the article and its additional files. The results of PESSA's built-in data analysis are

## Abstract

The activation levels of biologically significant gene sets are emerging tumor molecular markers and play an irreplaceable role in the tumor research field; however, web-based tools for prognostic analyses using it as a tumor molecular marker remain scarce. We developed a web-based tool PESSA for survival analysis using gene set activation levels. All data analyses were implemented via R. Activation levels of The Molecular Signatures Database (MSigDB) gene sets were assessed using the single sample gene set enrichment analysis (ssGSEA) method based on data from the Gene Expression Omnibus (GEO), The Cancer Genome Atlas (TCGA), The European Genome-phenome Archive (EGA) and supplementary tables of articles. PESSA was used to perform median and optimal cut-off dichotomous grouping of ssGSEA scores for each dataset, relying on the survival and survminer packages for survival analysis and visualisation. PESSA is an open-access web tool for visualizing the results of tumor prognostic analyses using gene set activation levels. A total of 238 datasets from the GEO, TCGA, EGA, and supplementary tables of articles; covering 51 cancer types and 13 survival outcome types; and 13,434 tumor-related gene sets are obtained from MSigDB for pre-grouping. Users can obtain the results, including Kaplan–Meier analyses based on the median and optimal cut-off values and accompanying visualization plots and the Cox regression analyses of dichotomous and continuous variables, by selecting the gene set markers of interest. PESSA (https://smuonco.shinyapps.io/PESSA/ OR http://robinl-lab.com/PESSA) is a large-scale web-based tumor survival analysis tool covering a large amount of data that creatively uses predefined gene set activation levels as molecular markers of tumors.

available here: https://data.mendeley.com/datasets/ph3d6sn75t/1 (DOI:10.17632/ph3d6sn75t.1).

**Funding:** The author(s) received no specific funding for this work.

**Competing interests:** The authors have declared that no competing interests exist.

## Author summary

The activation levels of biologically significant gene sets are emerging tumor molecular markers and play an irreplaceable role in the tumor research field; however, web-based tools for prognostic analyses using it as a tumor molecular marker remain scarce.

PESSA is an open-access web tool for visualizing the results of tumor prognostic analyses using gene set activation levels. A total of 238 datasets from the GEO, TCGA, EGA, and supplementary tables of articles; covering 51 cancer types and 13 survival outcome types; and 13,434 tumor-related gene sets are obtained from MSigDB for pre-grouping. Users can obtain the results, including Kaplan–Meier analyses based on the median and optimal cut-off values and visualization plots and the Cox regression analyses of dichotomous and continuous variables, by selecting the gene set markers of interest.

Freely available on the web at https://smuonco.shinyapps.io/PESSA/ OR http://robinl-lab.com/PESSA. Website implemented in R Shiny, with major browsers supported.

## Introduction

Cancer is a major public health problem worldwide. A total of 609,820 people are predicted to die from cancer in the United States in 2023 [1]. To reduce the threat posed by cancer, medical researchers have been exploring potentially effective therapeutic approaches and technologies to improve patient benefit from targeted therapies and immunotherapies via biomolecular markers as emerging strategies [2–4]. The widespread use of biomolecular markers in targeted cancer therapies and immunotherapy has significantly improved the clinical benefits for patients, and mortality rates for serious malignancies such as lung cancer [5], kidney cancer [6], leukaemia [7], and melanoma [2] have declined dramatically compared with those observed in the past decade, bringing hope to both oncology patients and medical researchers [1].

Studies have revealed that signalling pathways play important roles in human physiology and pathology, especially in oncology, where abnormalities in signalling pathways are closely associated with cancer cell activities, including tumorigenesis, growth and proliferation, invasion and metastasis, and apoptosis [8–12]. Currently, the level of signalling pathway activity is widely used in tumor diagnosis [13], molecular classification [14], prognostic evaluation [15] and treatment [16]. The use of overall activation levels of signal transduction pathways/gene sets as molecular markers for tumors is an emerging trend building on monogenic tumor markers [17–20]. Comprehensive analysis of the expression of multiple related genes within a signal transduction pathway/gene set can more sensitively identify subtle–but indicative of obvious–consistent functional changes in the expression level of each gene, thus addressing the limitation that conclusions based on single-gene tumor markers may overlook the overall biological role played by the pathway containing the gene [21]. In a study by Cao et al. on human bladder cancer (BLCA), it was found that EMT-related genes may be valid prognostic candidates for BLCA after considering the results of gene set variation analysis (GSVA), gene set enrichment analysis (GSEA), and survival analyses [22]; then, based on risk scores, it was concluded that patients with low risk scores had higher survival rates than those with high risk scores, consistent with the findings of Wang et al [23]. On the other hand, compared with single-gene prediction, the overall activation level of the gene set as a marker has the advantages of having lower volatility and producing more stable prediction results [24]. Based on this observation, overall gene set activation levels calculated using pathway enrichment analysis have been widely used in the prediction and assessment of cancer occurrence, progression,

and prognosis, as well as in drug discovery and development [25] and treatment [24,26–28] This approach greatly assists clinicians and researchers in screening tumor patients, guiding the monitoring of tumor patients, and performing decision-making for personalized treatment modalities.

Currently, large publicly available databases such as The Cancer Genome Atlas (TCGA) [29] and Gene Expression Omnibus (GEO) [30] store large amounts of pancancer multiomics data. Among the included data are gene expression profiling data accompanied by detailed clinical data. However, the complexity of the database interfaces and the large number of datasets, which are difficult to select, make it difficult to fully exploit these data. On the other hand, the commonly used survival analysis methods and pathway enrichment analysis (e.g., over-representation analysis (ORA), gene set enrichment analysis (GSEA), and single-sample gene set enrichment analysis (ssGSEA)) for assessing the activation levels of gene sets are challenging for clinicians and researchers who lack a bioinformatics background [31]. Therefore, an accurate and easily accessible web-based analysis tool that provides the ability to screen tumor prognostic markers based on pathway enrichment analysis is urgently needed. Although many web tools for single-gene tumor prognostic marker exploration are available, there are a limited number of existing web tools that use the overall activation level of a gene set as a tumor marker, and these tools have some obvious limitations. The tool represented by the CeNet Omnibus application [32] breaks new ground by using a self-developed algorithm, liquid association [33], to assess the overall activation level of a gene set for prognostic analysis. However, this tool can only analyse the sequencing results of ceRNAs and miRNAs and does not cover the full range of RNA types. Moreover, tools such as ESurv [34] and BRCA-Pathway [35] use conventional Gene Ontology (GO) [36] or Kyoto Encyclopedia of Genes and Genomes (KEGG) [37] methods for pathway enrichment analysis. The ssGSEA algorithm, which is capable of performing pathway enrichment analyses on single samples with high confidence, has not been fully utilized. Finally, the above web tools either incorporate only some of the TCGA tumor datasets or support users only in uploading their data, thus failing to fully exploit the rich resources in the public databases. Thus, a web tool that incorporates pancancer transcriptome data from multiple large databases and predicts tumor markers from pathway activation levels calculated by the ssGSEA algorithm would more effectively meet the needs of oncology researchers and clinicians.

Therefore, we developed PESSA, a web tool for prognostic analysis based on the R Shiny framework that is easy to access by all biomedical scholars and provides pathway-enriched gene set activation levels as molecular markers for tumor prognosis. We plan to incorporate tumor transcriptome data from several large databases, covering a wide range of tumor types and survival outcomes. In PESSA, ssGSEA algorithms will be used to assess the level of pathway activation in individual samples, and multiple survival analysis methods and custom visualizations will be available on the associated web pages. We hope that PESSA will become a useful tool for oncology researchers and clinicians.

## Materials and methods

### Transcriptome data collection

We collected human tumor-related transcriptome data from three large publicly available databases, GEO, TCGA, and EGA; and supplementary tables of articles. We obtained microarray datasets from GEO via GEOmirror (0.2.0) [38] and GEOquery (2.60.0) [31] and retrieved high-throughput sequencing datasets via direct download from the GEO webpage. We obtained the high-throughput sequencing dataset from TCGA via the UCSC Xena Browser. All of the above datasets had accompanying detailed clinical information and survival

information for the represented patients. We excluded datasets with incomplete clinical information related to survival outcome and survival time and those with a sample size that was too small (<30) and finally included 180 GEO (S1 Table), 33 TCGA (S2 Table), and a total of 25 immune checkpoint inhibitor therapy-related pathway datasets from sources such as EGA, GEO, supplementary tables of articles, and others (S3 Table) datasets. All expression profiling data were subjected to log2(exp+1) transformation and normalization (between-array normalization) via the normalizeBetweenArrays function in the limma package (3.48.3) [39]; this function is used for batch effect removal and between-sample normalization, which removes between-group differences. Standardization is adjusted from zero before conversion to a logarithmic scale to avoid missing values or large variance. Studies have shown that standardization in the median range allows for reconciliation between noise and bias [40]. For survival data, survival times were standardized to be in months. Samples missing corresponding survival information were removed before proceeding to the next step of the analysis.

## ssGSEA evaluation

The Molecular Signatures Database (MSigDB; https://www.gsea-msigdb.org/gsea/msigdb) [41] is one of the most widely used and comprehensive gene set databases for gene set enrichment analyses. Since the creation of MSigDB, the cancer-associated gene sets have been continuously updated and expanded, and MSigDB now includes >10,000 gene sets. These gene sets more completely represent a wider range of biological processes and diseases [42]. The R package msigdbr was used to obtain data on predefined gene sets in MSigDB, including hallmark gene sets (Hallmarks), conventional biological pathway gene sets (C2CP, which includes relevant gene sets sourced from the BioCarta, KEGG, Pathway Interaction Database, Reactome, and WikiPathways databases), and a GO database-sourced tumor-related gene set (C5GO). Furthermore, the R package GSVA [43] and GSEABase were used for pathway enrichment analysis of the obtained transcriptome sequencing data by the ssGSEA [44] method. Gene expression values for a given sample were normalized by sorting, and enrichment scores were generated using the empirical cumulative distribution function (ECDF) for the genes in the signature and the remaining genes. We finally converted the transcript expression levels into ssGSEA scores to derive the activation level of each gene set in individual samples of different types of tumors.

## Core functionality: Survival analysis

The ssGSEA score allows functional gene set activation levels to be used as molecular markers of tumor prognosis for survival analysis, and the results are presented to the user in an interactive visual format that is the core functionality of PESSA. First, the various gene set ssGSEA scores obtained by preprocessing each dataset are used for dichotomous grouping by the median (or optimal cut-off) value. We remove gene sets with extremely unbalanced groupings (sample size below 5% of the overall population or above 95% of the overall population under either grouping scheme) to avoid errors in the analysis caused by a sample size that is too small. The implementation of the optimal cut-off grouping in [45] is performed via the surv_-cutpoint function in the R package survminer [46] and specifies that the sample size under either grouping scheme is not less than 5% of the overall population. The realization of the optimal cutoff value grouping relies on the method of selecting the most significant p-value [45,47]. The implementation of the survival analysis is performed via the survfit function within the survival package [48], which performs a log-rank test of survival outcome under the different ssGSEA score pregrouping schemes. We also use the coxph function for single-factor Cox proportional hazards regression analysis. For single-factor Cox proportional hazards

regression analysis, we perform analyses based on dichotomization of the ssGSEA scores as described above and analyses based on the ssGSEA score as a continuous variable. In addition, hazard ratios (HRs), 95% confidence intervals (95% CIs), p values as well as the p-value of the Schoenfeld residual test of Cox analysis limited to continuous variables [49], based on the cox.zph function in survival package [48]; calculated under different grouping schemes and via different survival analysis methods are calculated accordingly. By integrating the above analyses, we provide a table of survival analysis results focusing on the activation level of each gene set in the respective dataset as well as interactive, visualized Kaplan–Meier survival analysis results, which describe and compare the distribution of survival times between different groups unlike regression analyses to assess the effects of risk factors.

## Application implementation

PESSA (https://smuonco.shinyapps.io/PESSA/ OR http://robinl-lab.com/PESSA) is powered by Shiny [50] and is available online via shinyapps.io. PESSA uses the tidyverse [51] family of R packages (e.g., dplyr [52], purrr [53], and tibble) [54]) to provide support for internal tabular data transformations, the DT package [55] to provide users with interactive data tables, and the survminer package [46] to present visualizations of survival analysis results. In addition, programming languages such as HTML5, CSS, and JavaScript are used to assist in web page construction and beautification. Fig 1 summarizes the PESSA build process.

## Results

### Data overview

PESSA is an open-access web tool for survival data visualization based on the activation levels of pathway-enriched gene sets as molecular markers of tumor prognosis. PESSA (1.0.0) currently incorporates a total of 6,360,737 records distributed across 180 GEO datasets, 33 TCGA datasets, and 25 datasets from GEO, EGA, article schedules for treatment with immunosuppressive agents, which cover data for 51 types of cancers and 13 different survival outcomes. A total of 13,434 tumor-related gene sets are obtained from MSigDB (S5 Table), and the transcriptome expression profile of each sample in the GEO, TCGA, and patients treated with immunosuppressants datasets is transformed based on the ssGSEA score.

Detailed information on all included datasets can be quickly queried in the DATA tab via an interactive table; the available data include cancer type, dataset platform, survival outcome, intervention method, and treatment details (drugs used, etc.), as well as links to the corresponding datasets. The results can be saved locally, including copying to the clipboard and saving locally in various formats such as CSV, Excel, or PDF or printing, for further research and analysis.

### Documentation

Users with any questions about functions, needs, or suggestions as well as uploading customized data for analysis can provide effective feedback by the message box under the About tab and sending an email to the technician's detailed email address. All comments and suggestions will be thoughtfully considered, and we will update PESSA accordingly to meet the needs of our users. Moreover, the updated records will be synchronized with the homepage and displayed on this page. Under the FAQ section, we provide detailed answers to user queries. For example, users can check the list of all predefined gene sets covered by PESSA and their source web addresses or detailed descriptions of the 13 survival outcomes. This functionality will more helpfully assist users in conducting oncology research with PESSA.

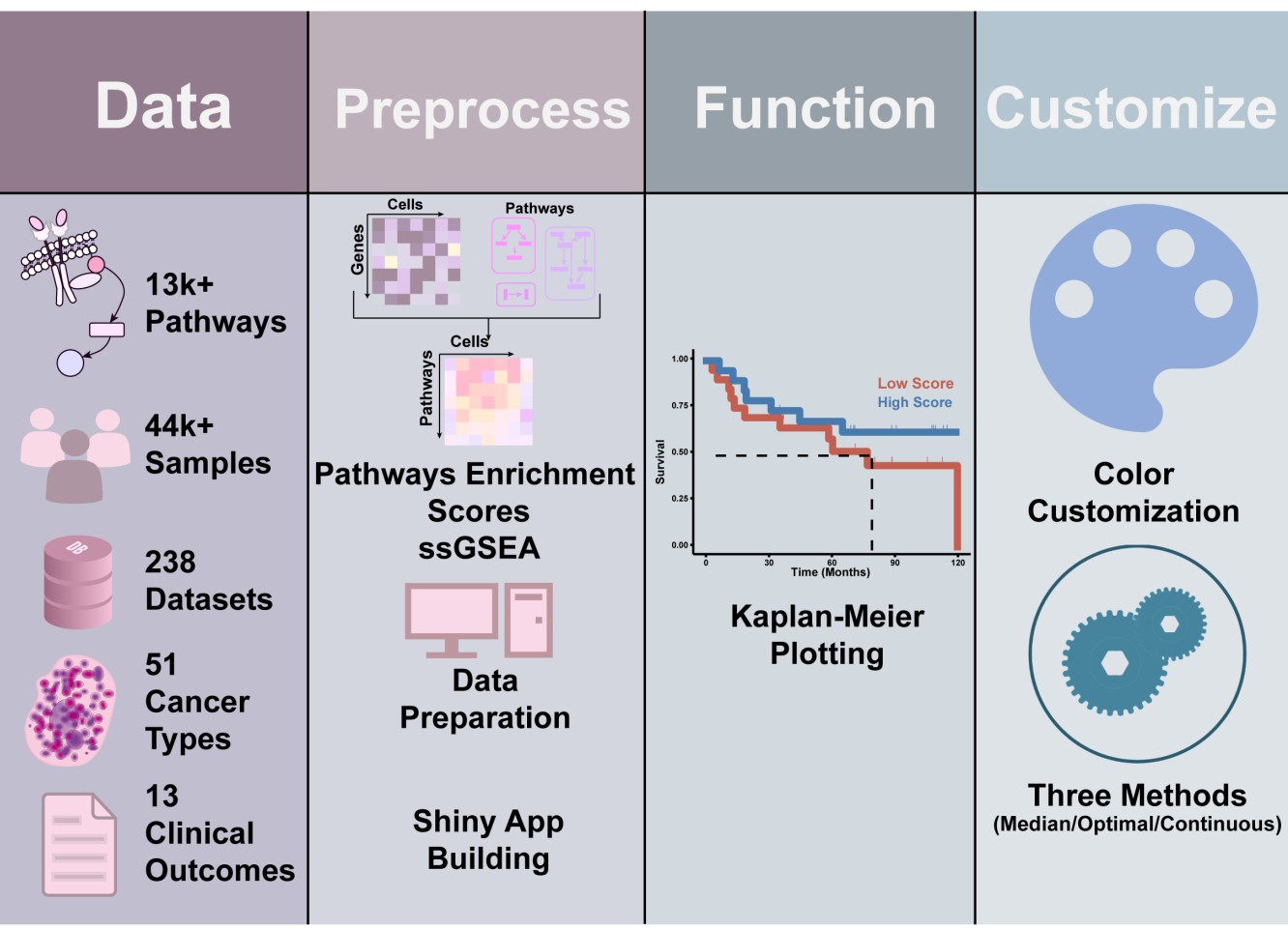

**Fig 1. PESSA workflow.** PESSA (v1.0.0) currently incorporates a total of 238 datasets from TCGA and GEO and a total of 13,434 predefined tumor-related gene sets from MSigDB. After data preprocessing in the background, PESSA provides the user with attractive and informative Kaplan–Meier curves and allows the user to customize the settings. Abbreviations: GEO: the Gene Expression Omnibus; MSigDB: the Molecular Signatures Database; TCGA: The Cancer Genome Atlas. Fig 1 cliparts are from Openclipart.

### Survival analysis

PESSA is a robust and powerful survival analysis tool that enables survival analysis using gene set activation levels in the pancancer domain by transforming transcriptome sequencing data into ssGSEA scores.

On the PLOT tab, PESSA guides the user through a three-part analysis process by interactively opening a collapsible box. First, the user selects/inputs the gene set of interest; second, based on the summary table of gene set-related survival analysis results displayed in the box, the user selects the target of interest and feeds it back into PESSA for visualization; and third, PESSA provides the user with interactively downloadable Kaplan–Meier plots that meet the requirements for submission to a wide range of journals.

An example of the use of PESSA is the exploitation of the activation level of the Reactome database-sourced gene set TGF-beta Receptor Activation Levels in Emt Epithelial to Mesenchymal Transition (TGF-beta Receptor Signaling in Emt Epithelial to Mesenchymal Transition), as a molecular marker for potential BLCA tumors. Survival analyses were performed using the TCGA-sourced BLCA dataset to analyse the survival outcomes of patients

dichotomized under the median ssGSEA score grouping scheme based on this pathway. The PESSA results showed that the overall survival (OS) time was significantly longer in the group with low TGF-β receptor EMT-related gene set activation levels than in the group with high corresponding levels, consistent with the results of Chen et al [56], who found that TGF-β-induced translation promotes BLCA metastasis by regulating epithelial-mesenchymal transition and invasive filopodia formation (Fig 2).

For the analysis after immunosuppressant treatment, the results obtained by PESSA were also as described previously [57]. We analyzed patients with human bladder cancer (BLCA) treated with PD1 inhibitors using HALLMARK—DNA REPAIR as the target DNA Damage Repair (DDR) pathway. The PESSA results demonstrated that patients with BLCA treated with PD1 inhibitors who exhibited a high degree of activation of the DDR pathway Further, we analyzed the PESSA melanoma dataset using PD1 inhibitors with the same results. (Fig 3)

## Discussion

PESSA is a large, interactive, and easily accessible web-based tool for clinical researchers to creatively analyse whether gene set activation levels can be used as a prognostic biomarker for cancer patients. PESSA provides a convenient opportunity for oncology researchers and clinicians to use the enormous amount of tumor transcriptome data available in GEO, TCGA, and other publicly available datasets without the need for programming skills, to annotate the predefined gene sets and ssGSEA scores in the MSigDB, to perform accurate survival analysis and to obtain attractive and informative Kaplan–Meier curves.

PESSA has advantages in many aspects, such as the breadth of predefined gene sets incorporated, the abundance of cancer types incorporated, the size of the dataset, the diversity of survival outcome information, the comprehensiveness of the analysis methods, and the user-friendliness of the customization functions. PESSA complements the scarcity of the previous survival analysis tools using the activation levels of predefined gene sets and innovatively provides a means for accurate and precise survival analysis by ssGSEA scoring of the genes that encompass the mRNA, miRNA, and long noncoding RNA (lncRNA) transcriptomes, thus expanding the data used by previous such analyses [32]. S4 Table shows a side-by-side comparison of PESSA with current common survival analysis visualization tools. It should be noted that although UCSC Xena has more advantages than PESSA in terms of data breadth and analysis function, PESSA is able to provide survival analysis based on specific gene sets, which is not available in UCSC Xena. Gene set-based analysis can help researchers to find specific pathways related to tumors, which can help researchers to discover specific biological functions of tumors from another angle. Moreover, ssGSEA has the feature of generating pathway activation scores for each sample, allowing for finer, personalized pathway associations with survival to explain heterogeneous activation between patients. Furthermore, compared to enrichment analysis methods such as GSEA [21],GSVA [43] and ORA [58], ssGSEA scores have been shown to better generalize known biology and validate experimental assays of pathway activity [44,59] as well as continuous ssGSEA scores allow for more robust survival modeling using the scores as a predictor variable as compared to dichotomous enrichment calls. The ssGSEA score has been proven by previous authors to be useful as a biomarker for survival prognosis in patients with tumors, which is helpful for tumor molecular classification [60] and treatment. For example, Zhang et al [61] applied the ssGSEA score to assess the relationship between autophagy and immune cell infiltration in hepatocellular carcinoma, innovatively suggesting that induction or inhibition of autophagy in combination with immunotherapy could be a prospective treatment strategy. Wei et al [62] suggested that autophagy inhibition could become an effective treatment strategy for hepatocellular carcinoma. Indeed, their study

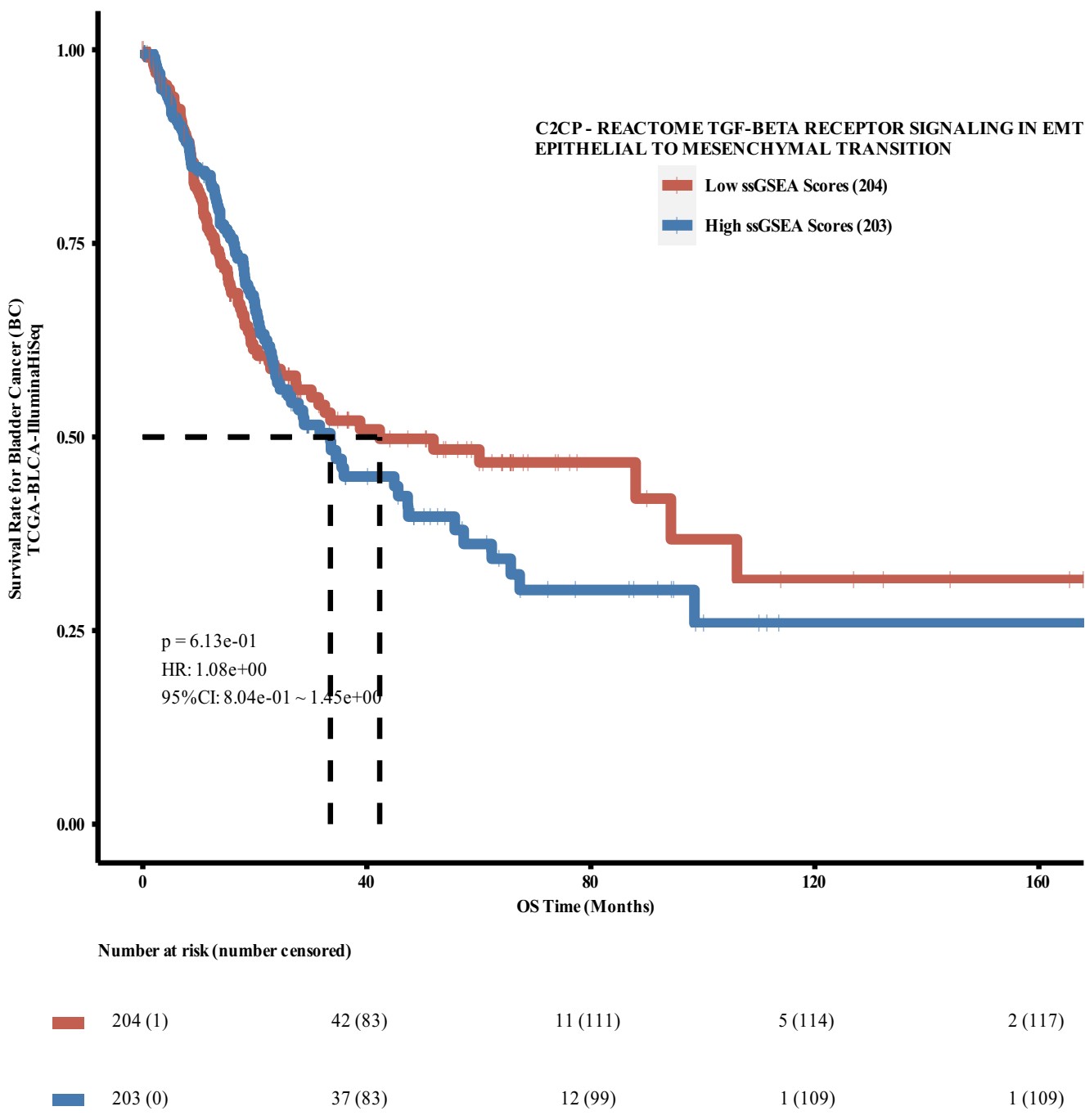

**Fig 2. Survival analysis feedback Kaplan–Meier curve.** Survival analysis of patients in the median-grouped TCGA-BLCA dataset using the gene set TGF-beta Receptor Activation Levels in Emt Epithelial to Mesenchymal Transition (TGF-beta Receptor Activation Levels in Emt Epithelial to Mesenchymal Transition) sourced from the Reactome database. The results from the TCGA-BLCA dataset indicate that BLCA patients with high levels of activation of the TGF-beta Receptor EMT-related gene set (TGF-beta Receptor Signalling in Emt Epithelial to Mesenchymal Transition) had significantly shorter OS times than patients with low levels of activation. Abbreviations: BLCA: Bladder Cancer; OS: Overall Survival; TCGA: The Cancer Genome Atlas.

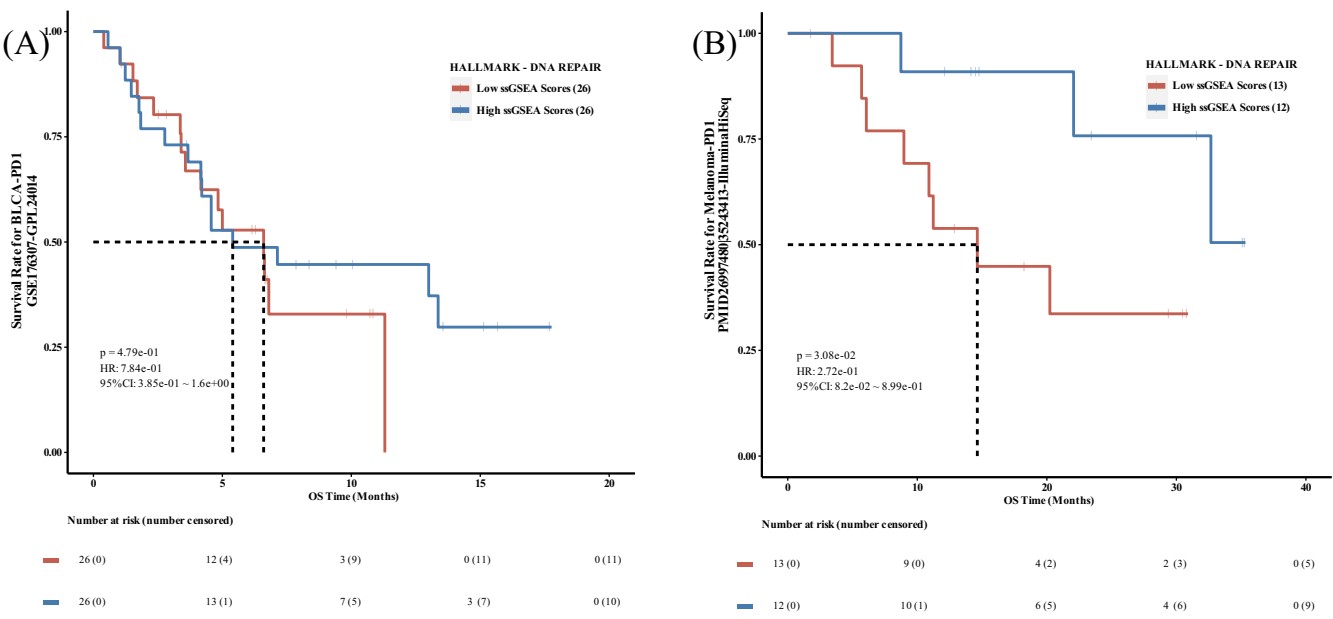

**Fig 3. Feedback Kaplan-Meier curves for survival analysis based on immunosuppressive therapy data.** A. Using the median dichotomous group-based dataset of patients with bladder cancer after PD1 inhibitor treatment, GSE176307, the results of the survival analysis demonstrated that the highly activated DNA damage repair pathway possessed a longer overall survival time. B. Examined using a built-in dataset derived from PMID 26997480 and 35243413 based on median dichotomous subgroups of PD1 inhibitor-treated melanoma patients, the results of the survival analyses demonstrated that the highly activated DNA damage repair pathway possessed a longer overall survival time.

demonstrated that the m7 regulatory factor-associated ssGSEA score was an independent prognostic factor for lung adenocarcinoma (LUAD) and liver hepatocellular carcinoma (LIHC) [62]. More importantly, regarding survival analysis methods, PESSA comprehensively provides Kaplan–Meier survival analyses based on subgrouping by the median and optimal cut-off values as well as regression analyses of the abovementioned subgroups and continuous variables with a one-factor Cox proportional hazards model. This is undoubtedly one of PESSA's major strengths. Moreover, the availability of tabulated results and the interactively customizable publisher-compliant Kaplan–Meier curve generator will satisfy the research needs of a wide range of oncology researchers and clinicians for analysing tumor prognosis.

Undeniably, there are still areas of improvement for PESSA, and we plan to complete these enhancements in subsequent releases. First, there are some limitations in the data used in PESSA. The original transcriptome sequencing data were derived only from microarray and high-throughput sequencing data in GEO, TCGA, EGA, and supplementary tables of articles; the predefined gene sets cover the commonly used gene sets related to tumors included in MSigDB. However, a large amount of potential target data–for example, data in supplementary infotmation of published oncology articles; the Sequence Read Archive (SRA), European Bio-informatics Institute (EBI), Surveillance, Epidemiology, and End Results (SEER), and other databases; the continuously updated GEO database; and the remaining predefined gene sets included in MSigDB–have not yet been incorporated in PESSA. We plan to update these data regularly in the future. Second, PESSA cannot analyse data uploaded by users, presenting an obstacle for users with unique samples, and it is thus likely to miss key discoveries in oncology. We are currently only able to encourage users to send data via the message board to our back office, which is processed and loaded into PESSA. We will endeavor to overcome this limitation in the future. Third, PESSA has not yet been able to satisfy the need for user-defined cut-

off values and currently can only use set cut-off values preprocessed in the background, a limitation that weakens the personalized features of PESSA. Fourth, due to the lack of information on tumor grading and treatment measures in some of the sample sets, PESSA is currently unable to satisfy subgroup analysis based on tumor grading, treatment measures, and tumor histology. This may cause the emergence of analysis heterogeneity to a certain extent, which affects the accuracy of the analysis. PESSA has currently demonstrated the treatment information under Data Tabs, and we will try to break through this limitation in the future.

Currently, the data available are still limited, and we plan to actively maintain and update PESSA, regularly replenish the database-derived data, and continuously improve the functionality of PESSA. In addition, our team plans to develop additional types of web tools for tumor survival analysis, including proteomics and epigenomics tools, to help researchers and clinicians in oncology to more comprehensively and efficiently explore the molecular level of tumors at the key junctures of possible diagnosis and treatment.

In summary, PESSA is an easy-to-access and easy-to-use analysis tool that can meet the needs of biomedical practitioners in exploring the survival prognoses of different cancer types and survival outcomes using specific predefined gene set activation levels as markers. We believe that PESSA will become a stable and reliable oncological tool with continuous improvement.

## Supporting information

**S1 Table. The 180 GEO datasets incorporated.**
(XLSX)

**S2 Table. The 33 TCGA datasets incorporated.**
(XLSX)

**S3 Table. The 25 ICIs datasets incorporated.**
(XLSX)

**S4 Table. A side-by-side comparison of PESSA with current common survival analysis visualization tools.**
(XLSX)

**S5 Table. Gene sets included in PESSA with corresponding brief description and detail pages.**
(XLSX)

## Author Contributions

**Conceptualization:** Quan Cheng, Kai Miao, Jian Zhang, Peng Luo.

**Formal analysis:** Hong Yang, Ying Shi, Anqi Lin.

**Resources:** Hong Yang, Ying Shi, Anqi Lin, Chang Qi.

**Supervision:** Quan Cheng, Kai Miao, Jian Zhang, Peng Luo.

**Visualization:** Hong Yang, Ying Shi, Anqi Lin.

**Writing – original draft:** Hong Yang, Ying Shi, Anqi Lin.

**Writing – review & editing:** Hong Yang, Ying Shi, Anqi Lin, Chang Qi, Zaoqu Liu, Quan Cheng, Kai Miao, Jian Zhang, Peng Luo.

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
