## [Decision Letter · Decision Letter 0]

19 Dec 2023

Dear Dr. Luo,

Thank you very much for submitting your manuscript "PESSA: A Shiny App for Pathway Enrichment Score-Based Survival Analysis in Cancer" for consideration at PLOS Computational Biology.

As with all papers reviewed by the journal, your manuscript was reviewed by members of the editorial board and by several independent reviewers. In light of the reviews (below this email), we would like to invite the resubmission of a significantly-revised version that takes into account the reviewers' comments.

We cannot make any decision about publication until we have seen the revised manuscript and your response to the reviewers' comments. Your revised manuscript is also likely to be sent to reviewers for further evaluation.

Sincerely,

Elena Papaleo, PhD

Academic Editor

PLOS Computational Biology

Jason Haugh

Section Editor

PLOS Computational Biology

Reviewer's Responses to Questions

**Comments to the Authors:**

Reviewer #1: Yang et al. presented PESSA, a shiny app for correlating pathway enrichment scores with survival outcomes. The team has calculated ssGSEA scores for the entire TCGA and 214 datasets from GEO, which covers 21 cancer types based on gene sets from MsigDB. The Shiny app provides an excellent interface for users to search pathways that might be associated with survival. The tool will be an excellent resource to perform high-level pan-cancer analysis. However, the current analysis neglects the heterogeneity of the patient cohort in each study, specifically related to histology, clinical grade, and treatment context, which can affect the interpretation of the results. There should be more effort to assess these covariates as additive effects and interactions within the systematic analysis via the CoxH model. Overall, the tool is of interest to the scientific community, but I would recommend the author to significantly expand its functionalities.

Major Comments.

1. The heterogeneity of the patient cohort needs to be included in the current resource. The patients need to be subsetted according to histology, clinical grade, and treatment context. I would also recommend incorporating these features as additive and interacting covariates in the CoxH model.

2. If Cox analysis is used, the author may include additional test statistics, such as the concordance index and chi-square test, to evaluate the proportion hazard assumption.

3. As the author states, the pathway tools might also be helpful in prioritizing gene features. It might be useful to perform a systematic screen for individual genes.

4. The provided MsigDB gene set is limited, and many of the immune-associated gene sets (also available to MsigDB) were not included. Given the significant interest in immunotherapy, the authors should also include results associated with these gene sets.

6. Since the current resource is designed for pan-cancer analysis, there should be an additional summary of pathways relevant in each histology, such as a) pathways implicated across squamous malignancies, b) pathways implicated in Herpes virus, c) pathways implicated in samples derived from metastatic events, d) pathways implicated in samples from pre or post immunotherapy treatment. There should be existing literature on the four studies, and it would be nice to provide examples in which PESSA was able to confirm or reject existing studies.

Minor comments:

1. The download feature is limited to the high-level summary with only 20-100 items that are viewable/downloadable at a time. I would recommend the author add the ability for the users to download the complete table.

2. Endpoints to the survival analysis should be defined, such as OS, PFS, DMFS, DFS, MFS, and BCR.

Reviewer #2: This manuscript describes a simple but useful tool to aid in analyzing gene set activation as a biomarker in cancer for prognosis. The tool is built as an R Shiny app that comes preloaded with many gene sets and expression data from a range of cancers. The app is easy to use and can provide statistical results and figures. Overall, I think the approach does have the potential to be of wide interest to cancer researchers. However, I do have some concerns.

Minor:

1. On line 63, there is a typo "820,2023"

2. On line 146, it is unclear what normalization was applied to arrays (which needs a citation) vs. sequencing data.

3. On line 174, it is not stated how the optimal cut-off is obtained.

4. There is no ability to control for confounding in models.

5. Gene sets are only given by name and there is no description, making them hard to use.

6. There is no ability to upload user data, which limites the utility of the tool.

Major:

7. The manuscript assumes that the overall method of using ssGSEA is useful compared to existing methods, of which there are many. However, it is not really established in either the background or the results.

8. Results from Cox proportional hazards models are given without the ability to check the assumptions. Making statistics easy does not make them correct.

9. In the app, for many gene sets, there are unreasonable HRs. For example, for C2CP - Biocarta Bad Pathway, in AML, with the IlluminaHiSeq platform, the HR is 8.16e+04 on the continuous Cox model. This is despite the fact that it is only 1.43 with the median cut-off. I suspect this is due to an issue with pre-processing the data, but there are other possibilities.

Reviewer #3: Benefits of Using PESSA

Gene Set Activation Levels

The authors argue that PESSA is a tool that distinguishes itself by focusing on gene set activation levels, offering a holistic view of pathway activities rather than individual gene expression levels. This approach can potentially provide more comprehensive insights into the functional status of biological pathways in the context of cancer.

The authors also state that PESSA shows robustness in pan-cancer analysis suggesting its effectiveness in handling diverse cancer types. The main feature is to analyze and compare survival outcomes across different cancer types, potentially revealing common or distinct pathway signatures.

Overall Considerations

PESSA may be particularly valuable for researchers interested in pathway-centric survival analysis across multiple cancer types, but KM Plotter, GEPIA, Prognoscan, and UCSC Xena offer a more extensive range of analyses, making them suitable for diverse research questions. The authors should be able to prove the need for a tool like PESSA and the benefit their tool provided against the specific functionalities and performances offered by each of the abovementioned tools KM Plotter, GEPIA, Prognoscan, and UCSC Xena.

In conclusion, the choice of genomic analysis tools depends on the specific goals and nature of the analyses researchers intend to conduct. PESSA's strengths in gene set activation and pancancer analysis are noteworthy, but authors must carefully consider the trade-offs in comparison to the broader functionalities offered by KM Plotter, GEPIA, Prognoscan, and UCSC Xena.

**Have the authors made all data and (if applicable) computational code underlying the findings in their manuscript fully available?**

Reviewer #1: **No: **Would recommend the author in providing the source code used to generate the result.

Reviewer #2: **No: **I don't see the code.

Reviewer #3: Yes

PLOS authors have the option to publish the peer review history of their article (what does this mean?). If published, this will include your full peer review and any attached files.

Reviewer #1: No

Reviewer #2: No

Reviewer #3: No
---

## [Decision Letter · Decision Letter 1]

19 Mar 2024

Dear Dr. Luo,

Thank you very much for submitting your manuscript "PESSA: A Web Tool for Pathway Enrichment Score-Based Survival Analysis in Cancer" for consideration at PLOS Computational Biology. Your manuscript was reviewed by members of the editorial board and by two of the three previous reviewers. Based on the reviews, we are likely to accept this manuscript for publication, providing that you modify the manuscript according to the review recommendations.

Sincerely,

Jason M. Haugh

Section Editor

PLOS Computational Biology

Reviewer's Responses to Questions

**Comments to the Authors:**

Reviewer #1: The authors has addressed all my questions.

Reviewer #3: Based on Supplementary Table 4, it is apparent that UCSC Xena provides more advanced analytical features compared to PESSA. While both platforms are user-friendly, UCSC Xena is distinguished by its extensive functionality and depth of information.

Summary endpoints from Supp.Table 4.

UCSC Xena: It aggregates data from over 1500 datasets covering 50+ cancer types, with nearly 85,000 samples. This extensive dataset coverage encompasses various genomic and clinical data types, providing researchers with a wealth of information for survival analysis.

UCSC Xena:

Supports Cox analysis, enabling researchers to assess the relationship between variables and survival outcomes.

Provides subgroup analysis functionality, allowing for the exploration of survival differences based on factors such as tumor grading, treatment measures, and tumor histology.

PESSA: has 233 datasets covering 51 cancer types, with over 44,000 samples. While still significant, it falls short in terms of dataset size compared to UCSC Xena.

Does not provide subgroup analysis based on tumor grading, treatment measures, and tumor histology, which may hinder the exploration of heterogeneity in survival outcomes.

The authors should highlight more on how their software offers a more thorough analysis of the different available online datasets, highlighting its advantages over UCSC Xena.

**Have the authors made all data and (if applicable) computational code underlying the findings in their manuscript fully available?**

Reviewer #1: **No: **The team should consider making the Shiny app source code available on zenodo prior to publication. This will ensure continuity of their online service when the shiny app is no longer available.

Reviewer #3: Yes

PLOS authors have the option to publish the peer review history of their article (what does this mean?). If published, this will include your full peer review and any attached files.

Reviewer #1: No

Reviewer #3: No

Figure Files:

Data Requirements:

Reproducibility:

References:

---

## [Editor Report · Decision Letter 2]

26 Mar 2024

Dear Dr. Luo,

We are pleased to inform you that your manuscript 'PESSA: A Web Tool for Pathway Enrichment Score-Based Survival Analysis in Cancer' has been provisionally accepted for publication in PLOS Computational Biology.

Best regards,

Jason M. Haugh

Section Editor

PLOS Computational Biology

---

## [Editor Report · Acceptance letter]

22 Apr 2024

PCOMPBIOL-D-23-01604R2 

PESSA: A Web Tool for Pathway Enrichment Score-Based Survival Analysis in Cancer

Dear Dr Luo,

I am pleased to inform you that your manuscript has been formally accepted for publication in PLOS Computational Biology. Your manuscript is now with our production department and you will be notified of the publication date in due course.

With kind regards,

Judit Kozma
